# Quasi-Solid-State Polymer Electrolyte Based on Electrospun Polyacrylonitrile/Polysilsesquioxane Composite Nanofiber Membrane for High-Performance Lithium Batteries

**DOI:** 10.3390/ma15217527

**Published:** 2022-10-27

**Authors:** Caiyuan Liu, Jiemei Hu, Yanan Zhu, Yonggang Yang, Yi Li, Qi-Hui Wu

**Affiliations:** 1Jiangsu Key Laboratory of Advanced Functional Polymer Design and Application, Department of Polymer Science and Engineering, College of Chemistry, Chemical Engineering and Materials Science, Soochow University, Suzhou 215123, China; 2Xiamen Key Lab of Marine Corrosion and Smart Protective Materials, College of Marine Equipment and Mechanical Engineering, Jimei University, Xiamen 361021, China

**Keywords:** electrospun polyacrylonitrile, polysilsesquioxane, separator, sol-gel preparation, quasi-solid-state polymer electrolyte, lithium batteries

## Abstract

Considering the safety problem that is caused by liquid electrolytes and Li dendrites for lithium batteries, a new quasi-solid-state polymer electrolyte technology is presented in this work. A layer of 1,4-phenylene bridged polysilsesquioxane (PSiO) is synthesized by a sol-gel way and coated on the electrospun polyacrylonitrile (PAN) nanofiber to prepare a PAN@PSiO nanofiber composite membrane, which is then used as a quasi-solid-state electrolyte scaffold as well as separator for lithium batteries (LBs). This composite membrane, consisting of the three-dimensional network architecture of the PAN nanofiber matrix and a mesoporous PSiO coating layer, exhibited a high electrolyte intake level (297 wt%) and excellent mechanical properties. The electrochemical analysis results indicate that the ionic conductivity of the PAN@PSiO-based quasi-solid-state electrolyte membrane is 1.58 × 10^−3^ S cm^−1^ at room temperature and the electrochemical stability window reaches 4.8 V. The optimization of the electrode and the composite membrane interface leads the LiFePO_4_|PAN@PSiO|Li full cell to show superior cycling (capacity of 137.6 mAh g^−1^ at 0.2 C after 160 cycles) and excellent rate performances.

## 1. Introduction

Solid-state electrolytes (SEs) exhibit outstanding features such as a large electrochemical potential window, suppressing lithium (Li) dendrites formation/growth and eliminating flammability, which have been believed as potential alternatives to replace the liquid electrolytes, particularly for high-safety lithium batteries (LBs) which are applied in electrical vehicles [1,2,3,4]. Various types of solid electrolytes have been developed, which mainly include inorganic Li-ion conductors such as ceramics, solid polymer electrolytes (SPEs) including polyethylene oxide (PEO), polyacrylonitrile (PAN), polyvinylidene difluoride (PVDF), etc., and quasi-solid-state polymer electrolytes (QSPEs) [5,6]. Ceramics conductors have high ionic conductivity at room temperature, but they are inherently fragile and brittle, and they are limited by the high interfacial resistance [7,8]. The SPEs have a low amount of interface contact, excellent processability and rich chemical adjustability, however they are easy to crystallize at room temperature, resulting in poor Li-ion conductivity and a low Li-ion transfer number [9,10,11]. One effective way to improve the Li-ion conductivity of the SPEs is to incorporate inorganic particles such as TiO_2_, SiO_2_, and Al_2_O_3_ into the polymer matrix to produce the polymer/ceramics composites (organic-inorganic hybrid materials) [12,13]. Recently, polysilsesquioxane-based SPEs have attracted lots of attention [14,15,16,17,18]. As a kind of special organic-inorganic hybrid material, the organic and inorganic components in polysilsesquioxanes are connected by segments. Therefore, they exhibit high mechanical strength which comes from the rigid segments and high ionic conductivity, which comes from the flexible segments simultaneously. The reported polysilsesquioxanes are usually prepared via a complicated in situ radical polymerization process, during which the flexibility of the electrolyte will be reduced, and the side reactions are easily produced [16,17]. In our previous work, we reported two self-standing ionogel electrolyte membranes that are derived from polysilsesquioxanes, which were prepared by a facile sol-gel transcription method which was followed by a sedimentation process, and these exhibited high Li-ion conductivity [18]. However, the membranes are composed of mutually entangled and stacked pure polysilsesquioxanes nanofibers, therefore, they lack structural stability.

The SPEs can also act as the separator between the cathode and anode in LBs [19]. Commercialized polyolefin membranes have many shortcomings, such as poor thermal and dimensional stability, poor wettability and low porosity (<50%), which will lead to a high level of internal resistance, low ionic conductivity, Li dendrite growth and even short circuits occurring [20,21,22]. Great effort has been devoted to improving the performance of the polyolefin membrane [23,24,25], such as the surface modification of it [26,27,28] or developing various diaphragms from other sources such as PAN, PVDF and polyurethane [29,30,31]. Especially, electrospun PAN microporous diaphragms for LBs have aroused lots of interest [32]. Compared with the traditional Celgard film, the electrospun PAN diaphragm has a higher degree of porosity, better electrolyte wettability, and higher ionic conductivity. Thus, the batteries using the electrospun PAN diaphragm showed superior cycling and better rate performances than those using the Celgard films [33,34,35,36]. However, the mechanical strength of the low molecular weight electrospun PAN film is unsatisfactory because of the weak adhesion between the fibers [37]. Moreover, the pure electrospun PAN nanofiber membrane cannot form a stable interaction between the electrolyte and the fiber when it is under pressure [38]. Mixing electrospun PAN with inorganic fillers or other polymers to prepare composite membranes is an effective way to solve these problems [39,40,41,42]. Nevertheless, the introduction of the inorganic or organic fillers into the polymer matrix may also cause agglomeration problems, which further decreases the interface compatibility between the electrodes/electrolytes [43,44].

To solve the above-mentioned problems that exist in the pure polysilsesquixane nanofibers electrolyte and the PAN separator and improve the overall properties of them (as is listed in Appendix A), in this work, a polysilsesquioxane/PAN composite membrane was fabricated by using an electrospun PAN nanofiber film as the substrate, cetyl trimethyl ammonium bromide (CTAB) as cationic surfactant and 1,4-bis(triethoxysilyl)benzene (BTEB) as the precursor. A layer of 1,4-phenylene bridged polysilsesquioxane (PSiO) was coated onto the surfaces of the PAN nanofibers in a sol-gel transcription way. The PSiO coating did not change the three-dimensional network structure of the matrix fiber films, but it improved the surface roughness and the absorption rate of electrolyte. The PAN@PSiO composite nanofiber membrane was then used as the polymer scaffold to absorb the liquid electrolyte to form QSPE as well as the separator. The electrochemical tests disclosed that this QSPE membrane exhibited thermal stability, high ionic conductivity, and the capability of effectively suppressing the growth of the Li dendrites. The assembled LiFePO_4_|PAN@PSiO|Li full cell exhibited a superior cycle stability and an excellent rate performance.

## 2. Experimental Section

### 2.1. Materials

PAN (M_w_ = 150,000) and N,N-dimethylformamide (DMF, 99.9%) were bought from Macklin. LiFePO_4_ was purchased from Shanghai D&B Biological Science and Technology Co., Ltd. (Shanghai, China). BTEB (97%) was obtained from Aldrich. CTAB (98%) was supplied by Bide Pharmatech Ltd. (Shanghai, China). Ammonia water (NH_3_·H_2_O, 25 wt%), hydrochloric acid (HCl, 36 wt%) and anhydrous ethanol were bought from Sinopharm Chemical Reagent Co., Ltd. (Shanghai, China). Ethylene carbonate (EC) and diethylene carbonate (DEC) were delivered from Guotai Huarong New Chemical Materials Co., Ltd. (Suzhou, China). Celgard 2325 was purchased from Suzhou Suke Lean Instrument Co., Ltd. (Suzhou, China). 

### 2.2. Preparation of PAN@PSiO Composite Nanofibers Membrane

First, PAN powders (1 g) were dissolved in anhydrous DMF (6 mL), and then, the solution was stirred for 3 h at 70 °C, which was used to prepare PAN nanofibers by electrostatic spinning technology with applied potential of 8 kV, collection distance of 15 cm and inflow rate of 0.2 mL h^−1^. The produced nanofibers were collected and dried in a blast oven at 40 °C overnight.

CTAB (200 mg) was dissolved in a mixed solution comprising ethanol (0.03 mL) and deionized water (50 mL). PAN nanofibers (0.05 g) were then immersed into the solution by adding 1 mL of BTEB and 0.6 mL of concentrated ammonia water. The solution was stirred for 12 h at 40 °C to prepare PAN/1,4-phenylene bridged PSiO composite nanofibers membrane (denoted as PAN@PSiO). The preparation process is indicated in Figure 1.

### 2.3. Methods and Characterizations

Scanning electron microscopy (SEM) images were taken using a Hitachi 4800 instrument (v = 3.0 kV). Transmission electron microscopy (TEM) images were conducted using a FEI TecnaiG20 (v = 200 kV). Wide angle X-ray diffraction (WAXRD) patterns were collected using a X’Pert-Pro MPD X-ray diffractometer with Cu Kα radiation (λ = 0.154 nm). The Brunauer–Emmett–Teller (BET) specific surface area and BJH pore-size distribution were measured using a Tristar II 3020 M, Micromeritics. Thermogravimetric analysis (TGA) data were acquired using a Thermal Analysis TG/DTA 6300 instrument at a heating rate of 10 °C min^−1^ in nitrogen atmosphere.

### 2.4. Battery Assembly and Electrochemical Tests

An electrochemical workstation (CHI660E, CH Instruments) was used to conduct electrochemical tests, and LAND systems were used to perform galvanostatic charge/discharge tests. The electrochemical impedance was obtained in a frequency of 100 kHz–10 MHz with AC amplitude of 5 mV at different temperatures. Linear sweep voltammetry (LSV) was used to measure the electrochemical stability window. The potential was swept between open circuit voltage and 7.0 V (V vs. Li/Li^+^) with a scan rate of 5.0 mV s^−1^. The charge/discharge cycling data of the symmetrical Li|PAN@PSiO|Li batteries were collected at room temperature to exam the interfacial compatibility between PAN@PSiO membrane and Li metal electrode using a Land CT2001A cell tester.

To construct a LiFePO_4_/Li full battery, firstly, LiFePO_4_, PVDF binder and acetylene black (AB) with a mass ratio of 8:1:1 were mixed in N-methylpyrrolidone to form a slurry, which was then spread on an Al foil, dried overnight in vacuum and cut into circles with a diameter of 14 mm to make the cathode. The active material that was loaded on the cathode weighed 1.5 mg. A metallic Li foil was used as the counter electrode. The PAN@PSiO membrane was cut into circles with a diameter of 16 mm and used as the separator. The composition of the liquid electrolyte is 1.0 M LiPF_6_ dissolved in a mixed solvent of DEC and EC (1:1 by volume). The CR2016 coin-type cells were assembled in an Ar-filled glove box. Cyclic voltammetry (CV) and charge/discharge cycling measurements were conducted at potential between 2.5 and 4.2 V, and the rate performance of the batteries was measured at different current density with 1 C = 170 mAh g^−1^ at room temperature.

## 3. Results and Discussion

The morphology and structure of the obtained PAN nanofibers and the PAN@PSiO composite nanofibers were disclosed by SEM and TEM images (see Figure 1a–d and Appendix A). The electrospun PAN nanofibers have a smooth surface and a uniform size, with a length of tens of microns and a diameter of about 300 nm, respectively. The fibers stacked together to form a 3D network structure. After depositing a layer of 1,4-phenylene-bridged PSiO, the surface roughness of the PAN@PSiO nanofibers was increased as observed in Figure 1b. The thickness of the PSiO layer is about 50 nm as calculated from Figure 1c. The obtained PAN@PSiO composite nanofiber membrane (Figure 1e) has a smooth surface, a uniform texture, no cracks, a white, round sheet, and a self-supporting ability. The refractive indexes of PAN and PSiO are similar, which effectively reduces the scattering effect, so the film becomes translucent after absorbing the electrolyte (Figure 1f) [45].

The thermal stability of the diaphragm is an important factor to determine the working efficiency and service life of the batteries. Therefore, the thermal shrinkage behavior and the thermo-gravimetric analysis (TGA) of the nanofiber membranes were tested. The optical photos of the three separators (Celgard 2325, PAN and PAN@PSiO) before and after heating at 150 °C for 0.5 h are compared in Figure 2a,b. It can be seen that the Celgard membrane changes from white to transparent and it shows a severe contraction after it was heated, while both of the PAN and PAN@PSiO membranes still maintained their original morphology, indicating the good structural stability of the prepared nanofiber membranes. The TGA curves (Figure 2c) show that almost no mass loss occurred before the temperature reached 300 °C for both of the PAN and PAN@PSiO membranes, indicating their excellent thermal stability.

Appendix A shows the WAXRD patterns of the PAN, PSiO, and PAN@PSiO samples. PAN has a characteristic peak at 2*θ* = 16.8°, corresponding to its unique crystal structure. PSiO displays a wide diffraction peak at 2*θ* = 23.1°, which is ascribed to its intramolecular siloxane structure. For PAN@PSiO, a sharp peak at 16.8° and a wide peak at 23.1° are both observed, which correspond well with the characteristic peaks of the two precursors, respectively. The above results indicate that PAN@PSiO has been successfully prepared, and PAN is in crystalline state, while PSiO in mainly amorphous.

Appendix A compares the stress–strain curves of the PAN nanofiber membrane and the PAN@PSiO composite nanofiber membrane. PAN shows a higher elongation at the break and a tensile strength of 0.9 MPa, which is because during the process of electrostatic spinning, the PAN molecular chain is directionally stretched and the interaction between the adjacent cyanide groups is destroyed, resulting in greater mechanical flexibility [38]. The tensile strength of the PAN@PSiO nanofiber composite membrane is increased to 1.24 MPa, and the addition of PSiO layer makes the film less flexible, leading to a decrease in the elongation at the break. As Zhang et al. reported [20] that generating SiO_2_ nanoparticles in a PAN solution could improve the performance of the electrospun PAN nanofiber membrane, whereas the agglomeration of SiO_2_ nanoparticles in the membrane reduced the mechanical strength of the membrane. In this work, the good compatibility between the organic segments in PSiO and PAN makes the PSiO evenly dispersed on the surface of the PAN fiber [46]. Moreover, the hard inorganic segments in PSiO improves the mechanical strength of the composite membrane, therefore the PAN@PSiO composite nanofiber membrane exhibits as good mechanical property as it is expected to do.

Appendix A shows the nitrogen adsorption–desorption isotherms and the BJH pore size distribution curve that were calculated from the adsorption branch for the PAN@PSiO composite nanofibers. A type IV curve with H3 hysteresis ring between the relative pressures (*P*/*P*_0_) of 0.44–1 can be observed in Appendix A, indicating that there are a large number of narrow and long pore structures in the sample, which may come from the stacking spaces between the PAN@PSiO composite nanofibers. The BET specific surface area and the relative pore volume of the PAN@PSiO composite nanofibers are 153.6 m^2^ g^−1^ and 0.16 cm^3^ g^−1^, respectively. In Appendix A, five peaks appear at the pore sizes of 2.9, 3.2, 3.5, 3.8, and 5.6 nm, confirming the existence of mesopores in the sample, which mainly come from the removal of the template in the PSiO layer. The PAN@PSiO composite nanofiber has abundant micropores and mesopores, providing enough spaces for the adsorption and storage of the electrolyte, which guarantees the rapid adsorption of electrolyte and high ion conduction [37].

Compared to the porosity and electrolyte intake of Celgard 2325, the PAN and PAN@PSiO membranes are displayed in Table 1. It can be found that the porosities of the PAN and PAN@PSiO membranes are much higher than that of the Celgard 2325 film, which provides them with a much higher electrolyte adsorption capacity. Moreover, the addition of PSiO onto PAN leads to a slight decrease in the porosity, however the Lewis acid–base interaction between PSiO and the electrolyte makes the electrolyte absorption ability of the composite membrane gently increase [20].

Ionic conductivity is an important index to balance the practicality of the polymer electrolytes and the separators. Therefore, the AC impedance test was carried out to calculate the value of the ionic conductivity of the prepared stainless steel (SS)|PAN@PSiO|SS symmetric cell. The AC impedance spectra and high frequency impedance spectra of the PAN@PSiO membrane at different temperatures are shown in Figure 3a,b, respectively. The intercept of the AC impedance spectra decreases in the high frequency region, and the radian of the spectra increases in the middle and low frequency region with the increase in the temperature, which occurred mainly due to the decrease in the charge transfer impedance and the interfacial impedance with the increase in the reactant activity. Figure 3c is the Arrhenius diagram of the ionic conductivity for the PAN@PSiO electrolyte membrane. With the increase in the temperature, the ionic conductivity of the polymer electrolyte increases continuously from 1.58 × 10^−3^ S cm^−1^ at 25 °C to 3.46 × 10^−3^ S cm^−1^ at 95 °C, which is due to the accelerated decomposition of the Li salts in the electrolytes and the faster migration of the Li ions as temperature rose. Using the SS sheet as the cathode electrode, the Li metal as the anode, the SS|Clegard2325|Li, SS|PAN|Li, and SS|PAN@PSiO|Li cells were assembled in the glove box that was filled with argon. As Figure 3d shows, the decomposition voltages of Celgard 2325, PAN, and the PAN@PSiO electrolyte membranes at room temperature are 4.6 V, 4.7 V, and 4.8 V, respectively, indicating that the nanofiber membranes possess excellent electrochemical stability.

The Li|PAN@PSiO|Li symmetric cell was fabricated in order to evaluate the interfacial stability of the quasi-solid-state electrolyte using the metallic Li electrode. The galvanostatic cycling test, which was charged/discharged periodically to mimic the Li stripping/plating process at room temperature at different constant current densities, is shown in Figure 4a,b. The rather stable voltages can be observed at the current densities of 0.05 mA cm^−2^ and 0.1 mA cm^−2^, which suggest the formation of stable interfaces and a good compatibility between the Li metal electrode and the PAN@PSiO electrolyte membrane. This result indicates that the PAN@PSiO electrolyte membrane has the good ability to depress the growth of the Li dendrites, as schematically shown in Figure 4c. The electrospun-prepared nanofiber membrane has good mechanical properties, which could efficaciously block the Li dendrites growth. After the Li stripping and plating experiments, the PAN@PSiO shows the integrated and connected structure as it is newly prepared, which can be found in Figure 4d.

To test the application of the PAN@PSiO electrolyte membrane in the full cell system, the LiFePO_4_|PAN@PSiO|Li battery was assembled using LiFePO_4_ as the cathode electrode and Li metal as the anode. The LiFePO_4_|Celgard 2325|Li and LiFePO_4_|PAN|Li batteries were also assembled for a comparison to be made. Figure 5a provides the CV curve of the LiFePO_4_|PAN@PSiO|Li battery at room temperature under a sweep speed of 0.1 mV s^−1^. A pair of good redox symmetry peaks appear at the potentials of 3.57 and 3.29 V, which suggests that the LiFePO_4_|PAN@PSiO|Li battery has good reversible embedding and embedded reactions. Figure 5b shows the 1st, 2nd, and 3rd potential-capacity cycling profiles of the LiFePO_4_|PAN@PSiO|Li battery at 0.1 C. The charge and discharge capacities of the first cycle are 165.4 mAh g^−1^ and 161.4 mAh g^−1^, respectively. The Coulombic efficiency is as high as 97.6%. The capacity loss that was caused by the SEI film on the Li metal surface in the first cycle is rather low. The potential of the charge–discharge platform is around 3.5 V (VS. Li/Li^+^), which is characteristic of a typical Li/LiFePO_4_ battery. The specific cycling capacity and the Coulombic efficiency of the three batteries at 0.2 C are shown in Figure 5c. All of the three batteries showed a high Coulombic efficiency and no significant capacity loss. The discharge capacity of the LiFePO_4_|PAN@PSiO|Li battery is 137.6 mAh g^−1^ after 160 cycles and it always kept the highest level of this during the process. Moreover, the rate performance of the batteries was also tested. Figure 5d compares the charge/discharge cycles of the three batteries at 0.1, 0.2, 0.5, 1.0, and 2.0 C. It is noteworthy that when the current density returned to 0.1 C, their specific discharge capacities were almost recovered, which indicates that the all of the batteries have excellent rate performances.

In order to study the electrochemical kinetic properties of the LiFePO_4_|PAN@PSiO|Li battery, AC impedance tests (at a current density of 2 C, frequency range 100 kHz ~ 10 MHz) were carried out before and after the cycling, the results of which are shown in Figure 6. Both of the spectra of the battery before and after the cycling show Nyquist characteristics. The intercept in the high frequency region represents the bulk impedance of the electrolyte, the semicircle in the middle and the high frequency region are associated with the interface impedance of the electrode and electrolyte and the SEI film, respectively, and the diagonal line in the low frequency region is related to the diffusion resistance of the Li-ions in the electrode. There is no significant change in the ontology impedance of the electrolyte before and after the circulation, indicating the excellent electrochemical stability of the LiFePO_4_|PAN@PSiO|Li battery. The interfacial impedance decreases from 368 Ω to 223 Ω, which is due to the improvement of the interfacial compatibility between the electrode and the electrolyte during the cycle.

The PAN@PSiO composite nanofiber membrane showed excellent mechanical, thermodynamic, and electrochemical properties, both as a quasi-solid-state electrolyte scaffold and as a separator for the LBs, which could be due to the following reasons: (1) a continuous network pore structure which was formed by three-dimensional stacked arrangement of nanofibers in the composite membrane provided the large porosity and the abundant Li ion transport channels; (2) the adhesion of the mesoporous PSiO layer on the PAN fiber surface increased the roughness and improved the mechanical strength and electrolyte uptake; (3) the organic-inorganic hybrid composition of PSiO enhanced the compatibility of the polymer composites. The synergistic effect of these factors made the composite membrane possess considerable structural stability and ionic conductivity.

## 4. Conclusions

In summary, a self-supporting PAN@PSiO composite nanofiber membrane was synthesized by a facile electrospinning and templating method. It was used both as a separator and a quasi-solid-state electrolyte scaffold. When it was compared with the pure PAN nanofiber membrane and the commercial Celgard 2325 separator, this PAN@PSiO membrane has improved its thermal and mechanical properties and had a higher absorption rate of the liquid electrolyte (297 wt%). The ionic conductivity of the prepared PAN@PSiO quasi-solid-state polymer electrolyte is 1.58 × 10^−3^ S cm^−1^ at room temperature, which is much higher than that of the traditional polymer electrolyte. Moreover, the PAN/PSiO composite nanofiber membrane has a three-dimensional structure, which could effectively block the growth of the lithium dendrites. These factors guarantee the LiFePO_4_|PAN@PSiO|Li full cell having an admirable cycle stability and an excellent rate performance. This work provides ideas for the research of new LBs with high-security and high-performance properties. In addition, the electrospinning matrix composite polymer electrolyte that was synthesized in this study can be prepared by other types of polymers or organosiloxane containing different functional groups, which provides a prospect for the development of a new generation of secondary batteries.

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
