# Peer review of "Quasi-Solid-State Polymer Electrolyte Based on Electrospun Polyacrylonitrile/Polysilsesquioxane Composite Nanofiber Membrane for High-Performance Lithium Batteries"

_materials, 2022, doi:10.3390/ma15217527_

Round 1

Reviewer 1 Report

In the manuscript "Quasi-Solid-State Polymer Electrolyte based on Electro-spun Polyacrylonitrile/ Polysilsesquioxane Composite Nanofibers Membrane for High-Performance Lithium Batteries," Authors present a new quasi-solid-state polymer electrolyte technology.

This article is interesting. However, there are some general points and some additional remarks in the text which need to be addressed before publication:

1.         The Introduction needs improvement. The Authors should emphasize why the current work is necessary.

2.         The Experimental section is poorly written; please add more detail about the experimental procedure and measuring apparatus used. The authors did not provide information about the purity of the used materials.

3. How many times have the experiments been performed? What is its repeatability? There are no error bars in the figures.

 The article is clearly written, and the illustrations are understandable. This manuscript will be interesting for all scientists involved in such studies, and I suggest publication after a major revision of the manuscript.

Author Response

Q1. The Introduction needs improvement. The Authors should emphasize why the current work is necessary.

A1. Thank you for your suggestion. We reorganized the introduction, and the revised part is highlighted in red. In the first paragraph, we raised the problem that pure polysilsesquioxane solid electrolyte lacked the mechanical strength. In the second paragraph, we proposed the agglomeration problem in PAN composites. Therefore, to solve these two problems, in this work, we prepared electrospun PAN/polysilsesquioxane composite nanofiber membrane, and they exhibited excellent propertied as expected.

Q2. The Experimental section is poorly written; please add more detail about the experimental procedure and measuring apparatus used. The authors did not provide information about the purity of the used materials.

A2. Thank you for your suggestion. The experimental section has been revised in sections 2.3 and 2.4, and the purity of the experimental materials has been marked.

Q3. How many times have the experiments been performed? What is its repeatability? There are no error bars in the figures.

A3. Thank you for your suggestion. The electrochemical experiments were usually carried out using three to four cells as parallel test. Since a cycle test lasted more than three months (even longer at lower current density), we couldn’t repeat it using the same cell. However, the similar cycling curves of parallel test could prove the reliability of the experiments. Besides, the ionic conductivity values were calculated from the AC impedance spectra, therefore, we didn’t add the error bars in the figures.

Reviewer 2 Report

The manuscript entitled “Quasi-Solid-State Polymer Electrolyte based on Electro-spun Polyacrylonitrile/Polysilsesquioxane Composite Nanofibers Membrane for High-Performance Lithium Batteries”. Some issues to be addressed will improve the quality of the manuscript. Therefore, I recommend this work could be published after the major revision

1.      Should the author write down the novelty of this paper?

2.      The English composition requires many improvements. The authors should proofread the manuscript carefully to minimize grammatical errors.

3.      All the references mentioned in the paper should be cited in the text or vice-versa.

4.      This research topic has been widely studied, and many studies have been performed. The author, please add a comparative table for the reader's clear understanding.

5.       The author should mention the mass of the working electrode. 

Reviewer 3 Report

The authors are advised to get help from a native English speaker to revise the manuscript.

Although aiming at producing more safe batteries, related to Li dendrite formation and growth and eliminating flammability, the work presented does not get to the point of testing those issues.

What actual novelties does this work convey to the Li-ion battery technology?

How do the results achieved compare to the best performant Li-ion batteries on the market? The comparison made in the conclusion section should be further extended.

Figures S2, S3, S4 and S6, which are in the supplementary documentation, should be taken to the main text. 

Author Response

Q1. The authors are advised to get help from a native English speaker to revise the manuscript.

A1. Thank you for your suggestion. The whole manuscript has been revised carefully to smooth the language and minimize the grammar and typo errors.

Q2. Although aiming at producing more safe batteries, related to Li dendrite formation and growth and eliminating flammability, the work presented does not get to the point of testing those issues.

A2. The potential profiles for Li plating/striping test as a function of time cycled at current density of 0.05 mA cm-2 and 0.1 mA cm-2 for the Li|PAN@PSiO|Li symmetric cell in Fig. 4a and 4b proved that the PAN@PSiO electrolyte membrane has the good ability to against the growth of Li dendrites. Fig. 4d further indicated the structural stability of the electrolyte membrane.

   The flammability of LBs is usually caused by the leakage of liquid electrolyte. In this work, the obtained quasi-solid-state electrolyte membrane constrained the electrolyte in the mesopores and spaces of PAN/PSiO nanofibers, therefore it can eliminate flammability from the source.

Q3. What actual novelties does this work convey to the Li-ion battery technology?

A3. The novelty of this paper is as follows.

1) the prepared PAN/PSiO composite nanofiber membrane showed excellent thermal, mechanical, and electrochemical properties compared with the commercially used Celgard 2325 separator;

2) the PAN/PSiO composite nanofiber membrane have three dimensional constructure, which could effectively block the growth of lithium dendrites;

3) due to the special organic-inorganic hybrid molecular composition and mesoporous structure of PSiO, the composite fiber membrane shows high electrolyte intake and good interfacial properties;

4) the PAN/PSiO electrolyte exhibits good ionic conductivity, which is as high as 1.58×10-3 S cm-1.

Q4. How do the results achieved compare to the best performant Li-ion batteries on the market? The comparison made in the conclusion section should be further extended.

A4. Thank you for your suggestion. The aim of this work is to solve the problems of pure PAN separator and polysilsesquioxane solid electrolyte, therefore we fabricated a novel electrospun PAN/mesoporous polysilsesquioxane composite electrolyte, which has not been reported before as far as we know. This composite membrane constructed a special organic/inorganic system. Though its performance still could not be compared with best performed Li-ion batteries on the market, it can be improved by following modifications such as tuning the composition of polysilsesquioxane and grafting functional groups.

   We compared some PAN- and polysilsesquioxane-based QSPEs in Table S1 in the supporting material.  

Q5. Figures S2, S3, S4 and S6, which are in the supplementary documentation, should be taken to the main text.

A5. Thank you for your suggestion. Figures S2, S3, S4 and S6 are listed in the supplementary documentation just because of the figure number limit in the main text.

Reviewer 4 Report

This manuscript proposes quasi-solid-state polymer electrolyte based on electrospun polyacrylonitrile/polysilsesquioxane composite nanofibers membrane for high-performance lithium batteries. The topic is interesting, and certainly consistent with the contents to be proposed to the readers of “Materials”. Moreover, the manuscript is well written and can be read with pleasure: this represents an important aspect in the current scenario of publications in international journals. Overall, I think that this manuscript has to be accepted, but the Authors should take into account the following minor revisions (in terms of bibliographic updates, grammar corrections and content deepening):

-          Detailed revisions: I spent several hours reading this manuscript, and Authors are asked to follow carefully the attached PDF file where I highlighted some points to be addressed. The attached file also contains language mistakes and typos; some questions related to manuscript contents could also be present and Authors must consider them properly before submitting the revised manuscript. A point-by-point reply is required when the revised files are submitted.

-          The Introduction should give a wider overview on the present scenario related to emerging matrices for polymer electrolytes, both in terms of recently published reviews and research articles. In particular, recent examples developed for Li and post-Li batteries are missing and a paragraph on this topic is highly suggested to be added in the Introduction. Authors are invited to go through the literature published in the last six months on these issues, and also on concepts developed some years ago in this field. Some of them are also mentioned in the above mentioned PDF file.

-          Authors should provide a clear explanation on the experimental error of the proposed research work. In particular, reproducibility of the phenomena described in the manuscript should be clearly stated in the “Results and Discussion” section; besides, some notes in the “Materials and Methods” section should be added highlighting which kind of experimental approach has been followed to check the reproducibility of the proposed system, the latter being of noteworthy importance in the present research field.

-          References: an article submitted to a journal should be consistent with the contents that it typically proposes in its table of contents. However, by checking the references of this manuscript, I did not find any articles published in this journal: this sounds rather strange. Maybe, Authors could check better the topics recently addressed by this journal, studying its table of contents and enriching the Introduction (as mentioned above) with some articles connected to this field.

Author Response

 Q1. Detailed revisions: I spent several hours reading this manuscript, and Authors are asked to follow carefully the attached PDF file where I highlighted some points to be addressed. The attached file also contains language mistakes and typos; some questions related to manuscript contents could also be present and Authors must consider them properly before submitting the revised manuscript. A point-by-point reply is required when the revised files are submitted.

A1. Thank you for your carefully review and kind suggestion. We revised the whole manuscript according to the attached PDF. All language mistakes and typos have been corrected.

Q2. The Introduction should give a wider overview on the present scenario related to emerging matrices for polymer electrolytes, both in terms of recently published reviews and research articles. In particular, recent examples developed for Li and post-Li batteries are missing and a paragraph on this topic is highly suggested to be added in the Introduction. Authors are invited to go through the literature published in the last six months on these issues, and also on concepts developed some years ago in this field. Some of them are also mentioned in the above mentioned PDF file.

A2. The introduction part in the text has been revised. The related samples have been compared in Table S1in the supporting material.

Q3. Authors should provide a clear explanation on the experimental error of the proposed research work. In particular, reproducibility of the phenomena described in the manuscript should be clearly stated in the “Results and Discussion” section; besides, some notes in the “Materials and Methods” section should be added highlighting which kind of experimental approach has been followed to check the reproducibility of the proposed system, the latter being of noteworthy importance in the present research field.

A3. The experimental section has been revised in sections 2.3 and 2.4. The electrochemical experiments were usually carried out using three to four cells as parallel test. Since a cycle test lasted more than three months (even longer at lower current density), we couldn’t repeat it using the same cell. However, the similar cycling curves of parallel test could prove the reliability of the experiments. Besides, the ionic conductivity values were calculated from the AC impedance spectra, therefore, we didn’t add the error bars in the figures.

Q4. References: an article submitted to a journal should be consistent with the contents that it typically proposes in its table of contents. However, by checking the references of this manuscript, I did not find any articles published in this journal: this sounds rather strange. Maybe, Authors could check better the topics recently addressed by this journal, studying its table of contents and enriching the Introduction (as mentioned above) with some articles connected to this field.

A4. Thank you for your suggestion. References 43 and 44 from Materials have been added in the text.

Reviewer 5 Report

The paper titled, ‘Quasi-Solid-State Polymer Electrolyte based on Electro-spun Polyacrylonitrile/Polysilsesquioxane Composite Nanofibers Membrane for High-Performance Lithium Batteries’ presents a new quasi-solid-state polymer electrolyte technology. PAN@PSiO nanofiber  composite membrane is proposed by using sol-gel method and electrospinning. Authors further claim its ionic conductivity as 1.58×10-3 S cm-1 in addition to excellent cyclic and rate performance. The carried out work is rigorous and contributes to development of separators for LiBs which is the need of current energy storage systems. However, I have following comments/suggestions which will further improve the manuscript, if incorporated.

1.      The abstract must contain quantitative results, as here only ionic conductivity is mentioned.

2.      Figure 1 needs much improvement, in terms of exact fabrication process. For example. Sol-gel method is missing here.

3.      I would recommend to add the optical photos of separators at 200 oC in figure 2. This characterization can be seen in [1-2], as they are most relevant works for the authors.

4.      There should be a dedicated figure which illustrates the fabrication process.  Figure 2 shows, however, no clear understanding can be taken by the reader.

5.      There are many grammatical mistakes which should be corrected.

6.      I would recommend adding a comparison table of similar works in the literature. This will add value.

7.      The quality of figures is vague/dull, it their resolution should be improved.

[1] http://dx.doi.org/10.1002/er.5371

[2] http://dx.doi.org/10.1149/1945-7111/abca71

Round 2

Reviewer 1 Report

The authors have fully commented on the comments. The article is suitable for publication in its current form.

Author Response

Thank you for your recommendation.

Reviewer 3 Report

Some improvement has been achieved in this new version of the manuscript. However, there persist many language flaws, namely verbal terms. Please check thoroughly these issues with the assistance of a native English speaker.
Most of the authors responses to my previous comments are relevant but they were not incorporated in the text, as they should have been.

Author Response

Q1: Some improvement has been achieved in this new version of the manuscript. However, there persist many language flaws, namely verbal terms. Please check thoroughly these issues with the assistance of a native English speaker.

A1: Thank you for your suggestion and sorry for our mistake. The text has been checked again carefully to improve the language.

Q2: Most of the authors responses to my previous comments are relevant but they were not incorporated in the text, as they should have been.

 A2: The manuscript has been revised again and the relevant responses to the following comments have been incorporated in the text and lighted in RED.

     Q3’: What actual novelties does this work convey to the Li-ion battery technology?

     A3’: In the Conclusion section, we emphasized the novelties of this work on the Li-ion technology.

 “Compared with the pure PAN nanofiber membrane and commercial Celgard 2325 separator, this PAN@PSiO membrane has improved thermal and mechanical properties and higher absorption rate of liquid electrolyte (297 wt%). The ionic conductivity of the prepared PAN@PSiO quasi-solid-state polymer electrolyte is 1.58 × 10-3 S cm-1 at room temperature, which is much higher than that of the traditional polymer electrolyte. Moreover, the PAN/PSiO composite nanofiber membrane has three-dimensional structure, which could effectively block the growth of lithium dendrites…This work provides ideas for the research of new LBs with high security and high performance.”

     Q5’: Figures S2, S3, S4 and S6, which are in the supplementary documentation, should be taken to the main text.

     A5’: Figures S2, S3 and S4 characterized the structural properties of the composite nanofiber membrane. Since the main text couldn’t insert too much figures, they are put in the supplementary document. Figure S6 has been added to the main text as Fig. 6.

Reviewer 5 Report

It should be accepted now.

Author Response

Thank you for your recommendation, and we have checked and revised the English language and style again as required.